# Temporal and Spatial Effects of Urbanization on Regional Thermal Comfort

**Yang Zhang** [1,2,†]**, Chao Zhang** [3,†]**, Kun Yang** [1,4]**, Zongqi Peng** [1,4]**, Linfeng Tang** [1,4]**, Haimei Duan** [1,4]**, Changhao Wu** [1,2] **and Yi Luo** [1,4,*]

1   GIS Technology Resarch Center of Resource and Environment in Western China, Ministry of Education, Yunnan Normal University, Kunming 650500, China; yangzhangzs@163.com (Y.Z.); yangkun@ynnu.edu.cn (K.Y.); pengzongqiynnu@163.com (Z.P.); tanglinfeng0940@163.com (L.T.); haimei_duan@163.com (H.D.); wuchanghaochn@gmail.com (C.W.)
2   School of Information Science and Technology, Yunnan Normal University, Kunming 650500, China
3   Heilongjiang Academy of Agricultural Sciences, Harbin 150038, China; zhangchaoynnu@163.com
4   Faculty of Geography, Yunnan Normal University, Kunming 650500, China
*   Correspondence: lysist@ynnu.edu.cn
†   These authors contributed equally to this work.

**Abstract:** Human urbanization has a great impact on the surface ecological environment, and few existing studies have explored the impact of urbanization on regional comfort on a long time scale. This study took Chenggong District, Kunming City, Yunnan Province, China, where urbanization was obvious, as the study area, and used the comfort evaluation model to evaluate the annual summer Discomfort Index (DI) in different periods of urbanization. Meanwhile, the impact strength of each factor characterizing human activities (Impervious surface, Gross National Product, and Total Population) on DI changes was analyzed, and the contribution rate of the main factors was quantified. The experimental results show that (1) over the past 20 years, under the background of the rapid economic development of Chenggong District, the annual average DI in summer showed an upward trend. The growth rate after the completion of University Town (2010–2020, Post-UT) was higher than that before the completion of University Town (2001–2005, Pre-UT). University Town was growing much faster than other regions. The monthly average DI changes were similar to the annual average changes. However, due to the movement of students in University Town during the summer vacation, the growth rate of DI in June was significantly higher than that in other months. (2) In terms of spatial changes, DI in the central and northwestern parts of Chenggong District increased significantly from 2001 to 2020. There were differences in the change rate before and after the completion of University Town. The area occupied by significant growth areas in June was much higher than in other months. It is proved that the economic and social development of Chenggong District would impact the regional human comfort, and the construction of University Town has aggravated the intensity of this impact. (3) In the during-UT, affected by the complex changes in land use types, the DI in Chenggong District showed fluctuations in time, but there was no obvious change in space. (4) The correlation and contribution analysis showed that the annual average DI in summer was closely related to human activities, especially the impervious surface had a strong contribution rate of 52.7%. The research shows that the development of new cities would have a strong impact on regional DI changes. And the obtained results will provide theoretical support for rational planning and management in the process of urban development in the future, thereby promoting the sustainable development of the region.

**Keywords:** contribution rate; discomfort index; human activity; urbanization

## 1. Introduction

Thermal environmental problems brought by the accelerated urbanization process will pose serious challenges to human living conditions [1–5]. During the development of

large cities, the change of the underlying surface and the influx of large numbers of people will cause heat to accumulate in the urban area [6], resulting in the heat island effect [7–10], which will affect the distribution and transfer of heat in the region. Ultimately, it will lead to the decline of urban comfort [11], which is not conducive to human production activities. Relevant studies have shown that the increase in temperature will increase the incidence of human respiratory diseases or other diseases [12], and changes in the thermal environment will threaten the health of urban residents. Therefore, evaluating urban comfort is one of the important issues that must be considered in regional development and planning [13–15]. Meanwhile, thermal environmental problems have not only been found in the rapidly developing urban centers but also in their surrounding areas, especially in the new urban areas built under the guidance of policies [1]. Compared with the urban center with perfect urbanization, the newly-built area develops late, but the construction period is short and concentrated. The regional impervious surface will expand rapidly in a short time [16], and the population density will continue to increase, which will affect the urban heat island effect and surface temperature [17], thus interfering with the stable development of the urban thermal environment [18–20]. Finally, it will interfere with the change of urban comfort. Paying attention to the changing characteristics and influencing factors of comfort in new urban areas will help the rational layout of regional development [21].

The thermal comfort index plays an important role in evaluating urban comfort, and different index levels represent different subjective thermal perceptions. Existing studies have developed metrics to assess human thermal response to environmental conditions, such as Effective Temperature (ET) [22], Physiological Equivalent Temperature (PET) [23], Discomfort Index (DI) [24], etc. Among them, DI is an index that uses the current temperature and relative humidity to evaluate the crowd's discomfort with the surrounding environment and the health risk of life. Generally speaking, when the air temperature is between 18 °C and 23 °C, and the relative humidity is between 35% and 70%, the human body has the highest degree of adaptation to the environment [25]. The high temperature in summer leads to significant changes in thermal comfort. Therefore, the research on DI in summer can better understand the degree of human adaptation to the surrounding environment, and the research results can effectively improve the regional thermal environment. Yan, et al. [26] quantified the effect of vegetation change on air temperature. The results showed that the increase of vegetation could effectively improve the environmental problems caused by the increase in urban heat. Meanwhile, the urban heat island (UHI) effect and DI were negatively correlated with the vegetation area. When the vegetation coverage was greater than 55%, the urban thermal environment was stable. Da Silva, et al. [27] investigated UHI and DI in the Brazilian city of Campina Grande, and the results showed that urban DI varied with seasons. Mushore, et al. [28] investigated the effects of seasonal land cover changes on outdoor human thermal comfort in Harare, Zimbabwe, southern Africa. The results showed that outdoor thermal discomfort was higher in the hot season, and the comfort was strongest after rainfall.

With the expansion of the city, the heterogeneity within the region makes the DI have the characteristics of spatial variation. However, the DI obtained by the measured meteorological data of a single site cannot accurately describe the differences in thermal comfort within the region [29]. With the rapid development of aerospace technology, remote sensing images are widely used to explore the changing characteristics of geographic elements, such as land surface temperature (LST). Therefore, satellite remote sensing images provide a technical means to understand urban comfort on a spatio-temporal scale. At present, some studies have used satellite remote sensing technology to estimate the comfort of cities. Oroud [30] used digital elevation data, GIS tools, and other methods to generate a spatially continuous map of climatic elements for inferring thermal comfort levels at different spatial resolutions across Jordan. Feng, et al. [31] modified the temperature and humidity indices by using the surface temperature and normalized humidity indices retrieved from remote sensing images instead of the required air temperature and relative humidity, and analyzed the influence of landscape patterns on urban thermal comfort

from macro and micro levels. Mijani, et al. [32] proposed a least-squares adjustment (LSA) model based on principal component analysis (PCA) to model outdoor thermal comfort through remote sensing and climate datasets. Xu, et al. [33] combined remote sensing data with measured meteorological data to reveal the spatial details of urban comfort. The results show that areas with high building density significantly increased human thermal discomfort, while areas with high vegetation and water coverage reduced human thermal discomfort.

The above studies fully demonstrate the availability of remote sensing data to explore the changes in urban DI. However, existing studies (1) have mostly focused on the changing law of urban comfort in short time scales, but do not pay attention to the changing characteristics of long time series, and cannot well explore the influence of the urbanization process on human discomfort. (2) These studies have given more attention to the relationship between DI and meteorological factors and land use types. However, affected by urbanization, with the rapid development of the regional economy and the continuous influx of floating population, the impervious surface also expands rapidly, resulting in the continuous increase of environmental problems such as the UHI effect, which will lead to changes in regional DI. Therefore, it is very important to explore the relationship between human activities and DI [34]. (3) Most studies have focused on the large cities with developed economies and complete urbanization process while ignoring the effect of thermal environment issues on newly built cities, especially new urban areas established under the guidance of policies [35–37]. The initial urbanization level of the new urban area was low, but the economy developed rapidly under conditions of government policy support, which created favorable conditions for studying the impact of the urbanization process on regional DI [38]. Analyzing the change characteristics of regional DI and its influencing factors before, during, and after the establishment of the city will help the rational layout and sustainable development of the city. On this basis, this paper selected Chenggong District, Kunming City, China, as the study area, combined with MODIS remote sensing images, and used statistical analysis methods to construct a comfort evaluation system. This study explored the temporal and spatial variation characteristics of regional comfort before, during, and after the city's construction in summer. At the same time, due to the rapid changes in land use types and the large influx of population in University Town, DI changed rapidly. Therefore, this paper also explored the differences in DI changes in University Town (UT) and other areas in Chenggong, analyzed the impact of urban comfort with factors representing human activities, and quantified its contribution intensity.

## 2. Study Area and Data

### 2.1. Study Area

Kunming City in Yunnan Province is located in the subtropical monsoon climate zone and is known as the "Spring City". Affected by the altitude, the temperature is suitable all year round, and the annual average temperature in Kunming has been about 15 °C over the years. The average annual precipitation is about 1000 mm, the sunshine is sufficient, and the mean annual sun exposure is about 2250 h [19]. Chenggong District belongs to Kunming City and is located at 102°45′–103°00′ east longitude and 24°42–25°00′ north latitude, with an area of about 499 km and an altitude between 1500 and 2800 m. Chenggong District is near Dianchi Lake to the west and Yangzonghai Lake to the east, with sufficient sunlight and a mild climate. Chenggong District is one of the seven urban Districts in Kunming City, with 10 streets and 65 communities under its jurisdiction. The urban and rural, industrial and mining, and residential land area is about 129.55 km$^2$ (2018), mainly located in the northwest and central regions. At the same time, it is one of the urban districts with the highest green coverage and the most abundant garden plants in Kunming. The forest area is about 149.31 km$^2$ (2018), mainly located in the eastern region. Since 2003, Chenggong District (Figure 1) has become a key area of Kunming's development and a hot spot for economic growth and urban modernization. In 2010, based on urbanization development, the construction tasks of nine universities in Yunnan Province were basically

completed (Yunnan University, Kunming University of Science and Technology, Yunnan Normal University, Yunnan Nationalities University, Yunnan College of Traditional Chinese Medicine, Kunming Medical University, Yunnan Academy of Arts, Yunnan Transportation Vocational and Technical College, Yunnan Radio and Television University). The University Town (UT) is located in the Wujiaying Sub-District area of Chenggong District. The north is the seat of the Kunming Municipal Government, with rapid economic development. The east is connected to entertainment areas such as parks and orchards, and the environmental protection is good. So far, UT is the most important part of the population of Chenggong New Area, which is strongly influenced by periods of time associated with college students' activities such as class and vacation.

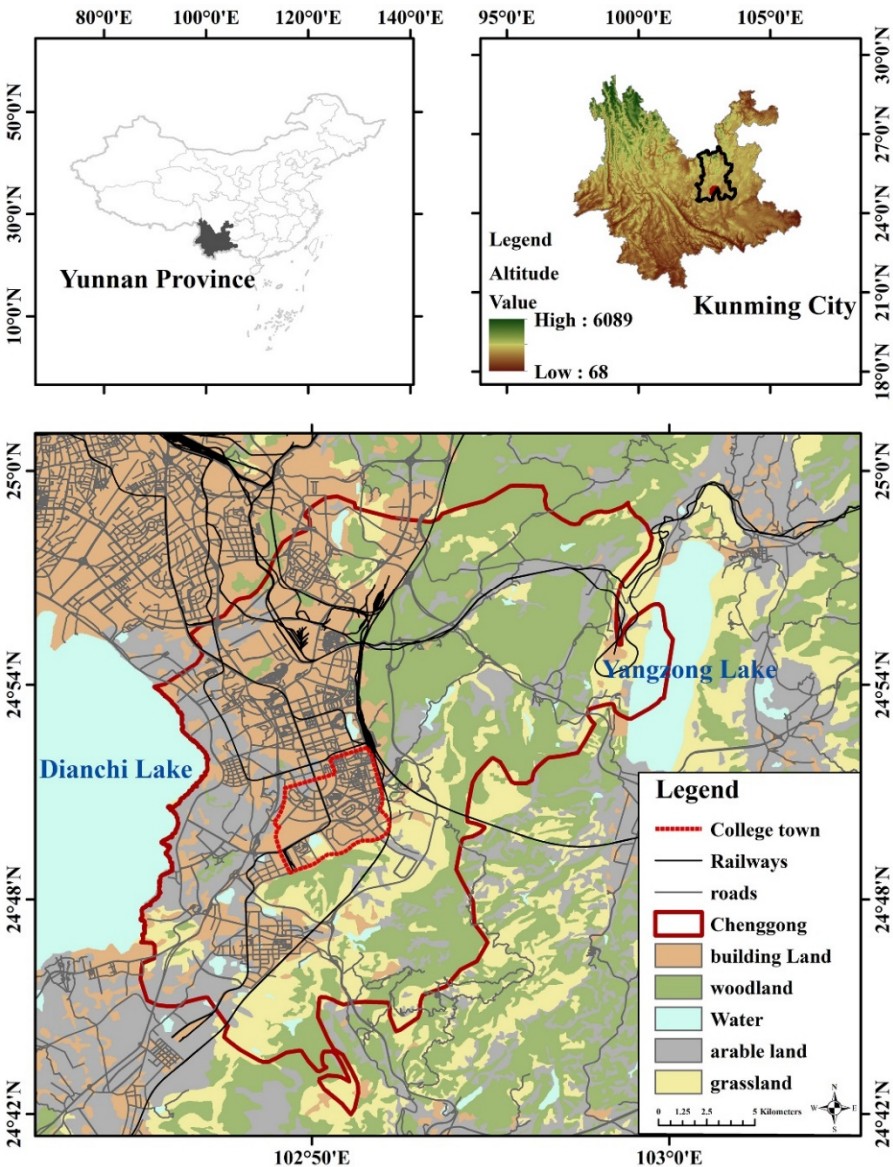

**Figure 1.** Study area. Note: The administrative boundary data in the figure comes from the National Geomatics Center of China (http://www.ngcc.cn/ngcc/, access on 21 March 2022). The land use data comes from the Resource Environment Science and Data Center (https://www.resdc.cn/, access on 21 March 2022) 30 m land use data for 2018. Road and track data comes from OpenStreetMap at https://www.openstreetmap.org/ (access on 26 October 2021).

*2.2. Data Sources*

The data for this research included land surface temperature (LST), relative humidity (RH), administrative boundary, and data representing human activities (impervious

surface area, IA, gross national product, GDP and population, POP). At the same time, the measured data of the thermal environment monitoring stations from June 2019 to December 2020 were also included to construct an air temperature data set and verify the accuracy of relative humidity. Among them, the LST data set came from MODIS's third-level product—MOD11A2, which is carried on Terra and Aqua satellites, and the data can be downloaded from NASA (https://ladsweb.modaps.eosdis.nasa.gov/, access on 30 October 2021). In this study, the regression equation between LST data and air temperature data was established to obtain air temperature data in the study area. For specific operation steps, please refer to Section 2.3. The relative humidity data came from the European Centre for Medium-Range Weather Forecasts (ECMWF), which were monthly averages. The IA data came from the impervious surface dataset of Gong, et al. [39]. After downloading, the administrative boundary of Chenggong District was used to mask and crop it, and the IA data set of Chenggong District from 2001 to 2018 was obtained by calculation. The GDP and POP data were from the statistical yearbook provided by the China Economy and Social Development Statistics Database (http://tongji.cnki.net/, access on 30 October 2021). The measured data came from 5 thermal environment monitoring stations set up in a medium-sized community in Chenggong District. Please refer to the "Methods" section for the site layout and related information.

### *2.3. Methods*

To better explore the characteristics and influencing factors of DI changes in Chenggong District from 2001 to 2020, the method of this study was mainly divided into three parts. The obtained data were used to calculate DI according to the comfort formula. Finally, the changing trend of DI was analyzed, and the influencing factors were analyzed using the Spearman correlation coefficient.

### 2.3.1. Air Temperature and Relative Humidity Data Acquisition

This study constructed the MODIS LST data set from 2001 to 2020, then established the relationship between the measured air temperature and the LST through the regression equation, and finally obtained the air temperature data set in Chenggong District according to the LST. For the relative humidity, this study used the monthly average relative humidity from the ECMWF and combined the measured data to verify its accuracy. The specific process is as follows:

(1) First, the MOD11A2 remote sensing image data product from 2001 to 2020 was downloaded. The downloaded images were reprojected to the WGS_84 coordinate system using the MODIS Reprojection Tool (MRT) software, resampled to a resolution of 1 km, and converted to GEOTIFF format for export. The data was then manipulated in ArcGIS 10.2 to set the cloud-affected vacancies to nulls. The resulting image was then mask-cropped using the Chenggong District boundary, and the missing values were interpolated. When the ratio of missing pixels to all pixels was less than 10%, the adjacent point interpolation method was used to interpolate the image; otherwise, the average value of the preceding and following eight days were used to interpolate the image [40]. Finally, the interpolated image was calculated, and the calculation was shown in Equation (1), where DN was the brightness value of the image pixel. After the calculation, the LST data set of the Chenggong area was obtained.

$$\text{LST} = \text{DN} \times 0.02 - 273.15 \tag{1}$$

(2) This study selected a medium-sized community in Chenggong District, and 5 thermal environment monitoring stations were arranged to record meteorological data. The monitoring frequency of the monitoring station is once per minute, which can accurately obtain meteorological data such as air temperature, relative humidity, wind speed, wind direction, solar radiation, and rainfall in the study area for continuous periods. Compared with intermittent data, continuous data can better highlight the

characteristics, trends, and laws of meteorological data in the study area. The thermal environment monitoring stations are scattered in 5 distant places, and the average value of the data of the 5 monitoring stations was taken as the verification data. Please refer to Table A1 for the specific parameters of the monitoring station. The monitoring time of the station was from 1 June 2019 to 31 December 2020. On this basis, this study used the measured air temperature and LST obtained in the previous step to perform regression analysis and thus obtained the regression equation (Equation (2)), whose coefficient of determination ($R^2$) was 0.857:

$$T_a = 0.8909\text{LST} + 5.4424 \quad R^2 = 0.857 \tag{2}$$

It can be seen from Figure 2 that there was a strong consistency between LST and air temperature, so the study used the above regression equation to calculate the air temperature.

(3) To verify the availability of relative humidity data, the study used the average relative humidity data collected by the monitoring stations and the relative humidity data obtained by the ECMWF for fitting analysis. The correlation coefficient r was 0.95, and through the significance test ($\alpha = 0.05$), it shows that the relative humidity in this paper can be used to calculate the change of regional DI.

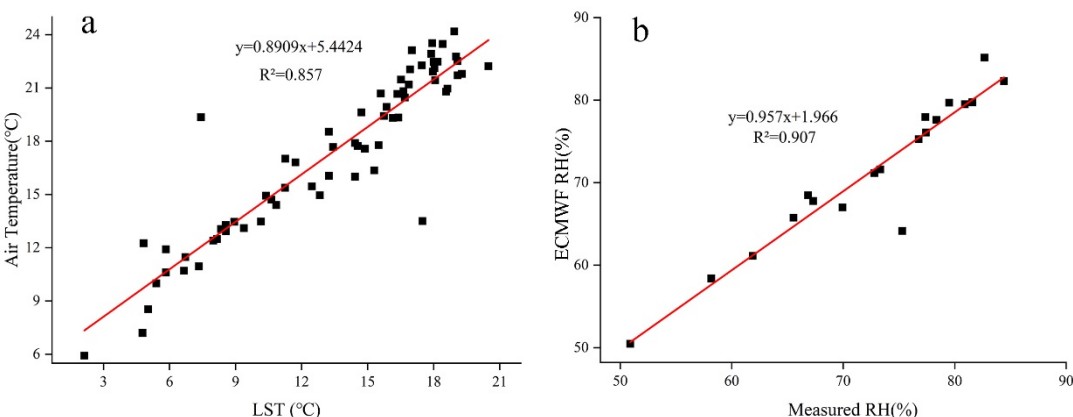

**Figure 2.** Regression analysis of air temperature and relative humidity. (**a**) The regression analysis of air temperature and LST; (**b**) The regression analysis of relative humidity data.

### 2.3.2. Discomfort Index (DI)

Researchers have used the Thom's Discomfort Index (DI) [24] to characterize the degree of human discomfort in the environment in summer, and found that it can better reflect human sensitivity to air temperature and relative humidity. At this point, the DI was calculated as a function of dry-bulb and loss-of-bulb temperature. With the development of research, researchers have improved the DI by using air temperature and relative humidity [41], and the calculation formula is as follows:

$$DI = T - 0.55(1 - 0.11RH)(T - 14.5) \tag{3}$$

In the formula, $T$ represents the space temperature and $RH$ represents relative humidity. The DI was divided into 6 levels to describe the comfort range (Table 1). When DI $\geq$ 32, it was regarded as a medical emergency. The smaller the DI, the higher the human comfort.

**Table 1.** Discomfort Index Classification Criteria.

| Level | DI (°C) | Uncomfortable Conditions |
|:---:|:---:|:---:|
| 1 | DI < 21 | No one feels uncomfortable |
| 2 | $21 \leq DI < 24$ | Less than 50% of people feel uncomfortable |
| 3 | $24 \leq DI < 27$ | More than 50% of people feel uncomfortable |
| 4 | $27 \leq DI < 29$ | Most people feel uncomfortable |
| 5 | $29 \leq DI < 32$ | Everyone feels serious pressure |
| 6 | $DI \geq 32$ | Medical emergency |

2.3.3. Correlation and Contribution Rate Analysis

To explore the influence of human activity factors on the change of DI in urban development, the study used the Spearman correlation coefficient to describe the relationship between the DI of Chenggong District and the three factors of IA, GDP, and POP. Since the IA data was up to 2018, this study only explored the correlation between human activity factors and DI from 2001 to 2018. In this study, the Spearman correlation coefficient was calculated by using SPSS 22.0 (Statistical Product and Service Solutions), and the calculation formula is as follows:

$$\rho = \frac{\sum_i (x_i - \overline{x})(y_i - \overline{y})}{\sqrt{\sum_i (x_i - \overline{x})^2 - \sum_i (y_i - \overline{y})^2}} \tag{4}$$

The Spearman correlation coefficient is often represented by the Greek letter $\rho$, the value range is $-1 \leq \rho \leq 1$, the $\rho$ value is greater than zero, each factor is positively correlated, and the $\rho$ value is less than zero, each factor is negatively correlated. The higher the absolute value of the $\rho$ value, the stronger the correlation between the two, and $x_i$, $y_i$ represent the ith variable of $x$ and $y$, respectively. On the basis of correlation analysis, this paper used Multiple regression analysis [42–44] to explore the contribution rate of main factors to the change of DI in Chenggong District. DI as a dependent variable is affected by the main factors. After establishing the multiple regression analysis equation, the research uses $R^2$ to describe the contribution of the main factors to the dependent variable. $R^2$ was a statistic to measure the fit of a model, as shown in Equation (5). *TSS* was the inherent variance of the response variable before performing regression analysis. *RSS* was the sum of squared residuals, indicating the variance that the regression model cannot explain. *IV* was the interpretable variance that the regression model could explain ($R^2 \in [0, 1]$). Regression analysis was also calculated by using SPSS 22.0.

$$R^2 = \frac{IV}{TSS} = 1 - \frac{RSS}{TSS} \tag{5}$$

**3. Results**

*3.1. Temporal Variation Characteristics of DI in Chenggong District from 2001 to 2020*

3.1.1. Overall Change Characteristics

This section analyzed the variation characteristics of DI in Chenggong District from 2001 to 2020 from the monthly and annual scales. As the most important part of the development plan of Chenggong District, the foundation of Chenggong UT was laid in 2005 and completed in 2010, and it had a strong impact on the regional thermal environment after completion. Therefore, the exploration of the change characteristics of DI was divided into three periods: before construction (2001–2005, pre-UT), during construction (2005–2010, during-UT), and after construction (2010–2020, post-UT).

From 2001 to 2020, the annual average DI in Chenggong District showed a significant upward trend, with a growth rate of 0.060 °C/a (Figure 3). The maximum value of annual average DI appeared in 2020 ($DI_{max}$ = 19.76 °C), the minimum value appeared in 2002 ($DI_{min}$ = 18.05 °C), and the difference between the maximum and minimum values was 1.71 °C. DI showed an upward trend before and after completion, but there were significant differences in the growth rate. The growth rate in the post-UT period was 0.091 °C/a, much

higher than that in the pre-UT period (0.043 °C/a) and higher than the overall growth rate from 2001 to 2020. During UT, DI showed a downward trend, and the change rate was −0.032 °C/a.

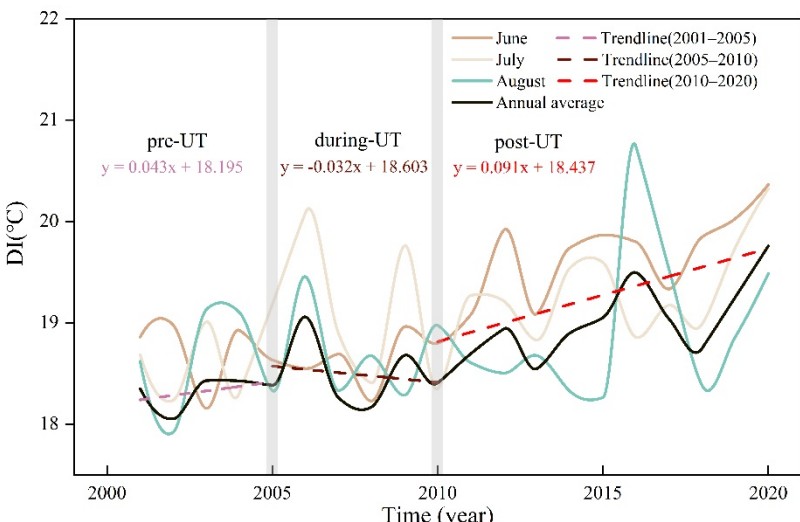

**Figure 3.** Annual and monthly changes of DI at different time scales in Chenggong District. Note: The red dotted line in the figure is the change trend of annual average DI, divided into two sections, before the completion of the university town (2001–2010) and after the completion of the university town (2010–2020).

In terms of monthly average DI, in the past 20 years, the DI in Chenggong District in June, July, and August all showed an upward trend, but the rate of increase was not the same. As can be seen from Figure 4, the annual average change rate of DI in June ($CR_{DI}$ = 0.094 °C/a) was the highest, which was higher than that in July ($CR_{DI}$ = 0.053 °C/a) and August ($CR_{DI}$ = 0.032 °C/a). The maximum value of DI in June appeared in 2020 ($DI_{max}$ = 20.19 °C), the minimum value appeared in 2003 ($DI_{min}$ = 17.83 °C), and the difference between the maximum value and the minimum value was 2.36 °C. The maximum value of DI in July appeared in 2020 ($DI_{max}$ = 20.00 °C), and the minimum value appeared in 2002 ($DI_{min}$ = 17.90 °C), and the difference between the maximum value and the minimum value was 2.10 °C. The maximum value of DI in August appeared in 2016 ($DI_{max}$ = 20.40 °C), and the minimum value appeared in 2002 ($DI_{min}$ = 17.56 °C), and the difference between the maximum and the minimum value was 2.84 °C. Separately, in the pre-UT period, except for June, all other months showed an upward trend. The average annual DI showed a downward trend in June, with a change rate of −0.051 °C/a. The change rate in July was the fastest, which was 0.107 °C/a, and the increasing rate in August was 0.061 °C/a. During UT, except for July, all other months showed an upward trend. The change rate in June was the fastest, which was 0.046 °C/a. Changes were flat in August, at a change rate of 0.001 °C/a. The annual DI showed a downward trend in July, with a change rate of −0.161 °C/a. In the post-UT period, except for July, the monthly change rates were higher than that in the pre-UT and during-UT. The average annual change rate of DI in June was the highest at 0.109 °C/a, the growth rate in July was 0.101 °C/a, and the growth rate in August was 0.067 °C/a. Compared with 2001–2005, the change of DI in June was the largest, increasing by 0.159 °C/a, the annual average DI in July decreased by 0.0565 °C/a, and the annual average DI in August increased by 0.006 °C/a. The average annual DI change in June is the most obvious, higher than that in July and August.

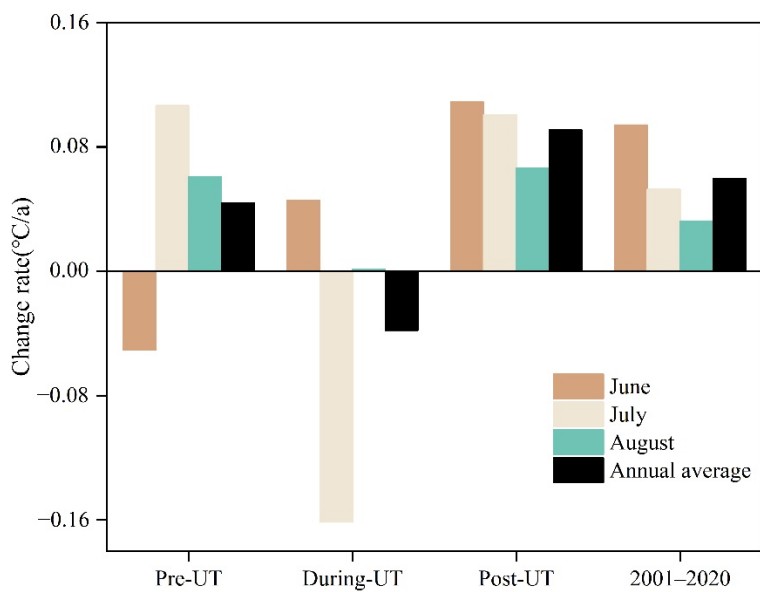

**Figure 4.** Average DI change rate in June, July, August, and year.

3.1.2. Change Characteristics of University Town and Other Areas in Chenggong District

In terms of annual average, from 2001 to 2020, the DI of UT and other regions showed an upward trend (Figure 5), and the growth rate of the UT (0.104 °C/a) was significantly higher than that of other regions (0.058 °C/a). Both regions showed an upward trend in the pre-UT and post-UT, and the growth rate of UT was higher than that of other regions. In the during-UT, DI had a downward trend, and the rate of decline in other regions was faster. In terms of the monthly average, the UT and other regions showed an upward trend in June, July, and August, but the growth rate of the UT was relatively high, 0.146 °C/a, 0.074 °C/a and 0.093 °C/a, respectively. The growth rates of other regions were 0.092 °C/a, 0.052 °C/a and 0.030 °C/a, respectively. University Town was on the rise both before and after construction. In the pre-UT, other regions showed a downward trend in June with a rate of −0.071 °C/a. Except for June, other regions showed a downward trend, and the decrease was fastest in July, with a change rate of −0.152 °C/a. Changes were flat in August. All other areas showed an upward trend in the post-UT, with the fastest growth in June, with a change rate of 0.115 °C/a.

*3.2. Spatial Variation Characteristics of DI in Chenggong District from 2001 to 2020*

Figure 6 shows the spatial distribution of DI in Chenggong District in the recent 20 years. It can be seen that the DI of Chenggong District was in the first and second levels (DI was less than 24 °C), and many areas were on the first level. From the overall spatial distribution, the DI in the west of Chenggong was significantly higher than in the East, and the regions with high values were mostly located in the northwest. With the time change, it can be seen that the high DI area (red area) not only increased but also continued to expand to the southeast. Separately, in the pre-UT period, DI was generally low. High DI areas were mainly located in the west and expanded slowly. Meanwhile, DI in University Town has been increasing, but its value is still low. During UT, the high DI region expanded rapidly in 2006, continued to shrink in 2007 and 2008, and expanded again in 2009. During this period, UT changed in line with the overall. In the post-UT period, the high DI areas accelerated their outward expansion and shifted from the northwest to the center. The expansion speed of the red area after completion was significantly higher than that before completion, and the high DI region accounted for the largest area in 2020. In UT, the high DI area increased significantly post-UT, and their values continued to increase over time. At the same time, there were many high DI in the lakeside areas of Dianchi Lake and Yangzonghai Lake, which may be due to the selection of MODIS nighttime data for the

study. The lake surface water temperature was higher than the LST at night, resulting in higher DI, so these abnormally high DI disturbed by other factors were not considered.

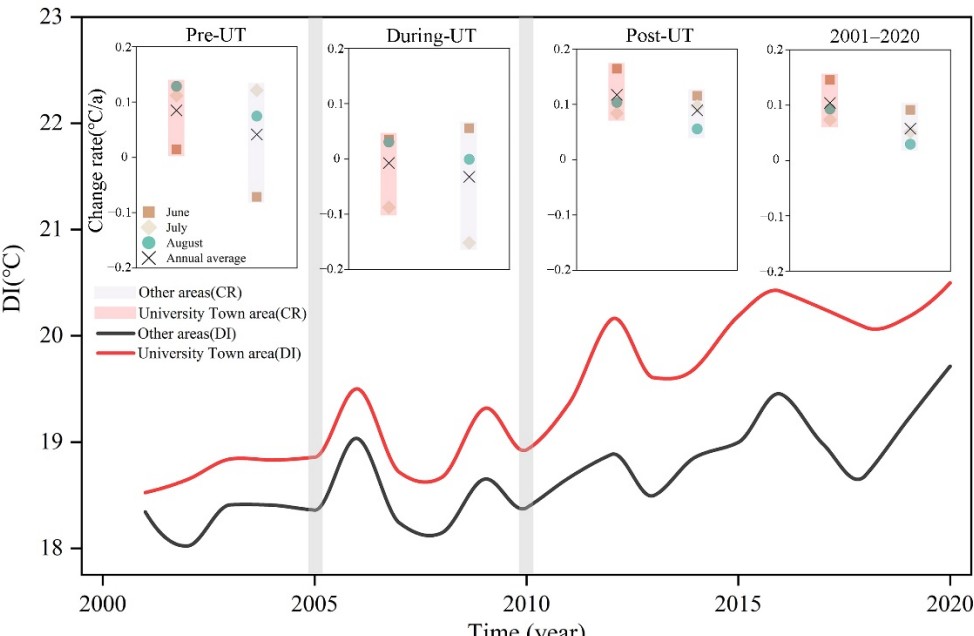

**Figure 5.** DI change trends in University Town and other areas in Chenggong.

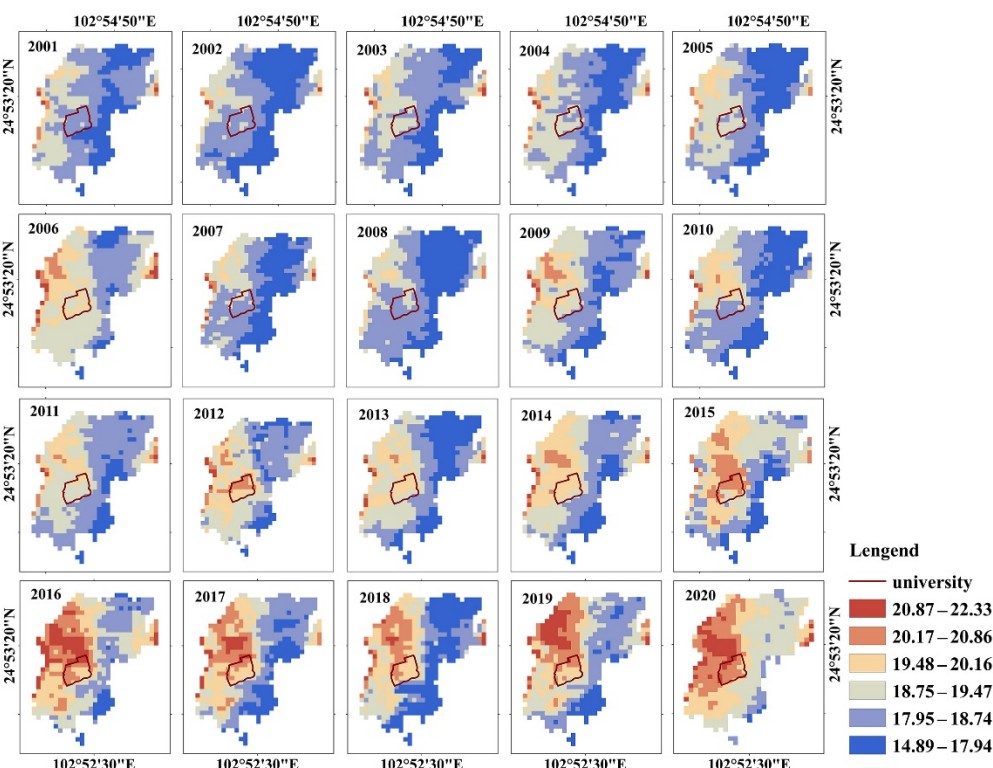

**Figure 6.** The spatial distribution of the annual average DI in Chenggong District from 2001 to 2020.

Figure 7 shows the spatial change rate of DI. For the spatial variation of the annual average DI from 2001 to 2020, there were few negative growth areas in Chenggong District. The northwest area showed a significant growth trend in patches, and the growth in other areas was relatively flat or almost unchanged. All areas of UT showed a significant upward

trend. In the pre-UT period, most of the areas did not change obviously, and the areas with significant growth rates were located in the central area of Chenggong District. For UT, the proportion of areas with no significant changes was higher. During UT, there was no significant change in DI in the Chenggong District. In the post-UT period, the significant growth area expanded compared with before construction, and it is mainly located in the northwest region. Meanwhile, the growth area of DI in the UT was more than before construction.

In terms of monthly averages (Figure 7a–c), between 2001 and 2020, the area proportion of DI decline in all months was low. In June, there were more areas with significant growth in DI than in other months in 2001–2020, and there was no area with a downward trend. All areas of University Town showed a significant upward trend. In the pre-UT, there was no significant change in most areas, and the area of DI reduction was mainly located in the northeast. There was no significant change in the UT area. During UT, the change trend in most regions was not significant. In the post-UT, the DI of Chenggong showed a significant growth trend in the northwest and central regions. College town had the highest percentage of DI growing regions of any month. In July, from 2001 to 2020, DI growth areas are discretely distributed in the north. DI in most areas of Chenggong District did not change significantly in the pre-UT and during-UT. Post-UT, some areas in the northwest were on the rise, but there was no significant change in UT. In August, from 2001 to 2020, the growth in the northwest and central regions was more significant, and some parts of the UT showed an upward trend. However, there was no significant change in the pre-UT, during-UT, and post-UT.

### 3.3. The Impact of Human Activities on DI Changes

The study selected three factors to characterize the intensity of human activities from 2001 to 2018, including the impervious surface area (IA), the population (POP), and the gross domestic product (GDP). Firstly, the correlation between three factors and DI on different time scales was obtained by using Spearman correlation coefficient analysis. Figure 8 shows the Spearman correlation coefficients for each factor and DI. Among them, the correlation coefficient between POP and average annual DI was the strongest, which was 0.695 ($p \leq 0.01$), the correlation coefficient between GDP and DI was 0.690 ($p \leq 0.01$), and the correlation coefficient between IA and DI was 0.691 ($p \leq 0.01$). The three factors were significantly positively correlated with annual DI. In June, July, and August, the correlation coefficient between IA and DI in June was the highest, which was 0.750 ($p \leq 0.01$), followed by GDP, and the correlation coefficient was 0.748 ($p \leq 0.01$). The correlation coefficient between POP and DI was 0.740 ($p \leq 0.01$). In June, the three factors were significantly positively correlated with DI. In July, the correlation coefficient between IA and DI was 0.346 ($p > 0.05$), the correlation coefficient between POP and DI was the strongest, which was 0.369 ($p > 0.05$), and the correlation coefficient between GDP and DI was 0.347 ($p > 0.05$). The correlation between the three factors and DI in July was weak, and none of them passed the significance test. In August, the correlation coefficient between IA and DI was the strongest, 0.150 ($p > 0.05$). The correlation coefficient between POP and DI was 0.131 ($p > 0.05$), and the correlation coefficient between GDP and DI was 0.148 ($p > 0.05$). The three factors had a weak correlation with the DI in August, and none of them passed the significance test.

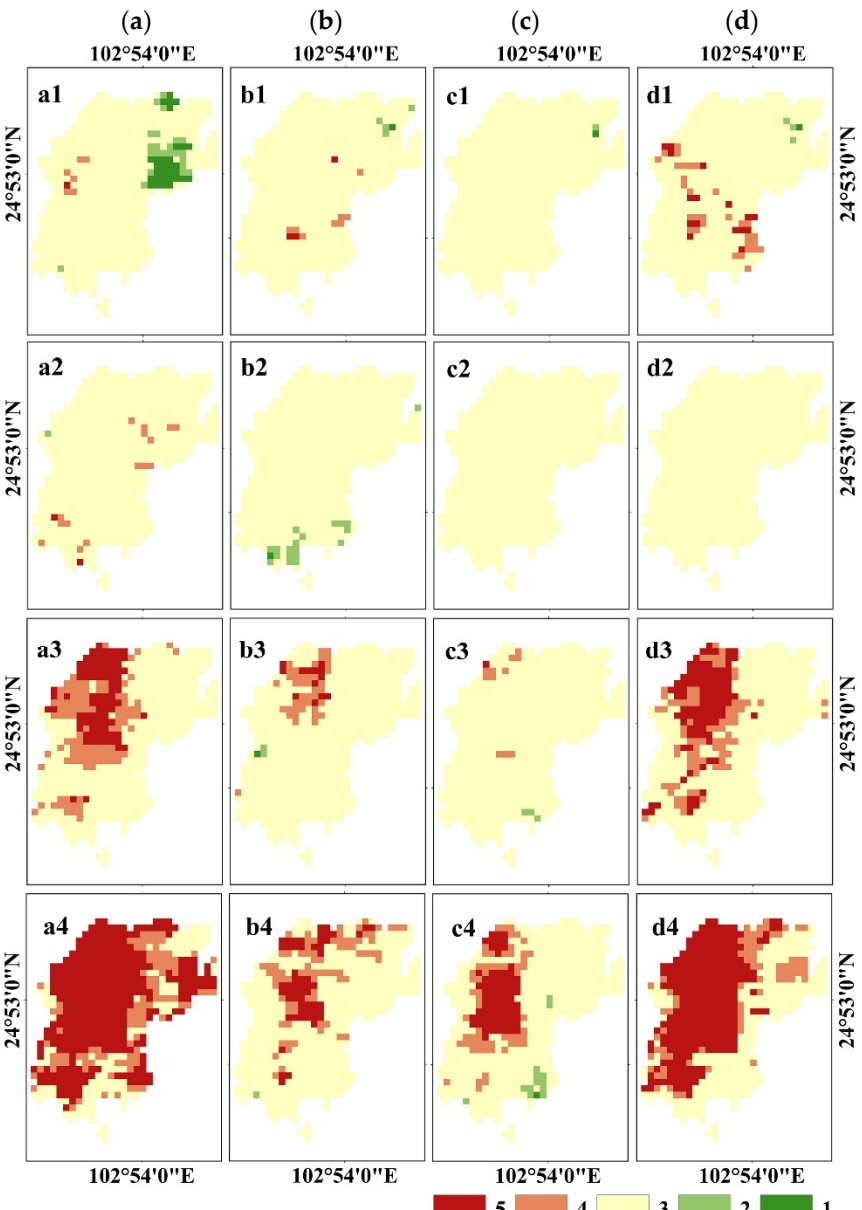

**Figure 7.** Spatial change rate of DI in Chenggong District from 2001 to 2020. Note: Colors 1–5 in the figure represent the trend of significant negative growth, negative growth, unchanged, positive growth, and significant positive growth, respectively. (**a**–**d**) represent the spatial change rate of DI in June, July, August, and the year, respectively. Figures 1–3 show the spatial change rate in the pre-UT period (2001–2010), in the post-UT period (2010–2020), and the overall (2001–2020).

After studying the correlation between human activity factors and DI changes in Chenggong District, it can be seen that the three factors were significantly correlated with the average annual DI changes, indicating that the DI was strongly affected by human activities. Therefore, the study used multiple regression analysis to quantify the contribution rate of the main factors to the DI change. In establishing the regression equation, it was necessary to discuss the multicollinearity between the factors. When one independent variable is related to another independent variable, it will lead to the inaccuracy of the calculation parameter estimation. Multicollinearity problems can be treated by the variance inflation factor (VIF). For those independent variables with VIF > 10, it is considered that there is a multicollinearity problem. If less, the multicollinearity is not significant enough and is ignored. The influencing factors of three variables meeting the conditions were subject to multiple linear regression with DI, and the results are shown in Table 2.

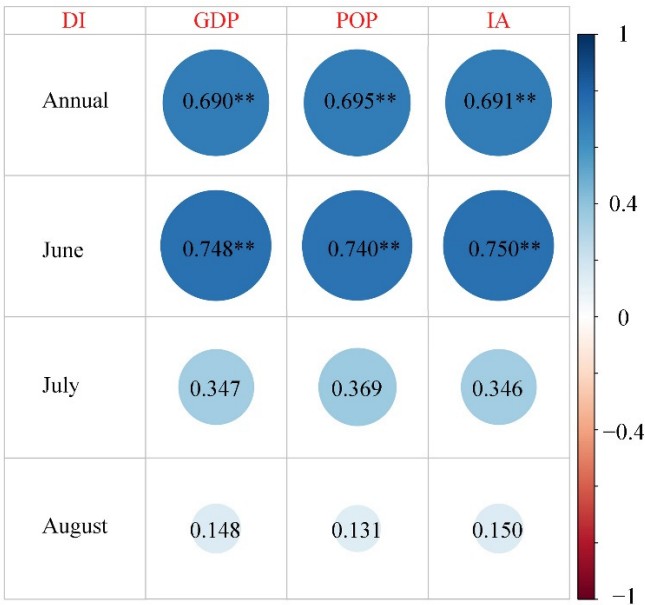

**Figure 8.** Correlations between human activity factors and DI (**: $p < 0.01$).

**Table 2.** Regression analysis results.

| Variable | Model Parameters | | | | Results |
|---|---|---|---|---|---|
| | **B** | **Beta** | **Sig.** | **VIF** | |
| IA | $5.838 \times 10^{-6}$ | 0.726 | 0.001 | 1.000 | R = 0.726 |
| POP | \ | −0.047 | 0.919 | 6.542 | $R^2 = 0.527$ |
| GDP | \ | −0.867 | 0.392 | 32.207 | F = 17.802 |

Among them, B is the regression coefficient, Sig represents the significant *p* value of the T test, VIF is the index of collinearity diagnosis, and R represents the goodness of fit of the prediction model. It can be seen from the results that the VIF value of GDP was greater than 10, and there was a collinearity problem, so the variable was removed. At the same time, in terms of significance, the significance of POP was 0.919, and the significance of GDP was 0.392, neither of which passed the significance test. Therefore, the two variables of GDP and POP were excluded from the construction process. Finally, the variable IA was used to analyze the impact of human activities on DI, and the regression equation was constructed. The results are as follows:

$$DI = 5.838 \times 10^{-6} IA + 18.108 \quad \left( R^2 = 0.527 \right) \tag{6}$$

The model results showed that IA was positively correlated with DI, and thermal discomfort increased gradually with increasing IA. $R^2$ was used to measure the degree of IA's interpretation of DI changes, which can show that in this study, from 2001 to 2018, IA contributed 52.7% to the DI change in Chenggong District.

## 4. Discussion

In this study, the air temperature and relative humidity data set in the Chenggong area was constructed through MODIS product data and the simulation data of ECMWF. The accuracy of the data set was verified through the measured data. The results showed that the calculated air temperature and relative humidity could meet the research needs. On this basis, the DI of Chenggong District from 2001 to 2020 was calculated. Firstly, through trend analysis, the study explored the temporal and spatial variation characteristics of summer DI in Chenggong District at different scales over the past 20 years. Then, using

the methods of correlation analysis and contribution rate analysis, the study explored the influence of the intensity of human activities on DI changes in the Chenggong District to better analyze the factors driving DI. The above analysis, combined with the results, is discussed as follows:

*4.1. Temporal and Spatial Variation Characteristics of Annual Average DI in Chenggong District*

In terms of temporal variation, from 2001 to 2020, the DI of Chenggong District showed an upward trend. In the post-UT period (2010–2020), the growth rate of DI was significantly higher than that in the pre-UT period (2001–2005) and higher than that in the past 20 years. It showed that the continuous construction of Chenggong District had promoted the rapid development of the regional economy and population growth, thus changing the thermal environment and affecting human comfort. With the accelerated development of the new area, DI was also growing. Especially after the construction of Chenggong University Town, college students have become the main part of the floating population. The increase in a large population and the construction of university campuses have changed the regional land use type and urban thermal environment, accelerating the rise of DI. The growth rate of the UT was significantly higher than that of other regions and the overall change in Chenggong District, which once again illustrated the impact of the construction of the UT on the regional thermal environment. During UT, Chenggong District, University Town, and other areas showed a downward trend. Through observation data, it was found that this was related to the significant increase in DI in 2006. The higher DI may be affected by the temperature, or it may be due to the large-scale construction of Chenggong University Town in 2006, which led to the conversion of grassland and forest land to bare land for buildings, resulting in a rapid increase in DI. At the same time, DI fluctuated greatly during construction, which may be due to the complex changes in land use, which have strong interference with DI.

From the spatial variation, the annual average DI in the western part of Chenggong District was higher than that in the eastern part, and the DI in the northwest and central parts was higher. Meanwhile, the area with high DI expanded over time. In the pre-UT period, DI was generally low, the high DI area expanded slowly, and DI in UT has been increasing, but its value was still low. In the during-UT, DI in Chenggong District fluctuated greatly. In the post-UT period, the high DI area has been continuously transferred from the northwest to the central area. In UT, the high DI area continued to increase over time. Secondly, the results of the annual DI spatial change rate showed that most areas of Chenggong District increased significantly from 2001 to 2020, and most of the growth areas were located in the western region. The area with a significant growth rate before the construction of the university city was lower than that after the construction. The above results showed that the change of annual DI in Chenggong District had strong spatial heterogeneity. Most of the areas with rapid economic development and rapid transformation of land use types in Chenggong District were in the northwest and central regions, which made the DI rise continuously. Most of the southeast regions were forest land and cultivated land, and the DI changed little. Especially affected by the construction of UT in 2010, the DI in the central region increased rapidly. With the construction of supporting infrastructure of University Town, the growth rate of DI after 2010 was higher than that before. Meanwhile, all areas of the UT showed a significant upward trend. During UT, the change of DI in Chenggong District was not significant, which may be related to the complex land-use changes during this period.

*4.2. Temporal and Spatial Variation Characteristics of Monthly Average DI in Chenggong District*

In terms of temporal variation, the DI of Chenggong District, University Town, and other areas showed upward trends in all months from 2001 to 2020. The growth rates of DI in June were significantly higher than that in other months. For Chenggong District, in the pre-UT period, DI showed a downward trend in June, an upward trend in July and August, and the fastest upward rate in July. In the post-UT period, all months showed an

upward trend, and the rising rate was significantly higher than that before completion and the overall change rate. Among them, June had the fastest rising rate and the largest change range. Meanwhile, after construction, the University City grew at a much higher rate in June than in other months and periods, and all regions showed significant growth trends from 2001 to 2020. This may be because the universities were on summer vacation in July and August, and students left school and returned home. A large number of population transfer changed the urban thermal environment, resulting in the slow rise rate of DI, which was far lower than that in June. The spatial variation characteristics of monthly average DI in Chenggong District also confirmed the impact of summer vacation students' leaving school on DI. From the spatial variation, in the past 20 years, the significant growth area of DI in all months of Chenggong District was mainly located in the central region. In June, the area of DI in Chenggong District increased significantly, which was much higher than that in other months. At the same time, the spatial changes were insignificant in all months before the completion but changed after. The construction of Chenggong UT had a strong impact on the DI changes. The action intensity was significantly higher in June than in other months, which was disturbed by the students leaving school and returning home during summer vacation.

*4.3. Study on Influencing Factors of DI Change in Chenggong District*

By analyzing the temporal and spatial variation characteristics of DI in Chenggong District from 2001 to 2020, it can be found that DI had a strong response to human activities. Therefore, the influencing factors of DI change were explored through correlation and contribution analysis. In terms of the annual average, the correlation analysis results showed that DI in Chenggong District had a significant positive correlation with IA, GDP, and POP and had the strongest correlation with POP. In terms of the monthly average DI, the correlation between the three factors representing human activities and DI was the highest in June. The correlation between DI and the three factors was weak in July and August. Secondly, the contribution rate results showed that the contribution rate of IA to DI change was high, which was 52.7%. It showed that the DI changes in the Chenggong area were strongly affected by human activities. The existing research has shown that the increase of impervious surface area would aggravate the urban thermal effect through the formation of surface thermal runoff [45], reduce the degree of human comfort, and promote the rapid growth of regional DI. This effect was also disturbed by the population [17]. During the summer vacation, the correlation between human activities and DI decreased in July and August, and its effect on human comfort decreased.

**5. Conclusions**

This study took Chenggong District, a new urban area of Kunming, as the research area, combined with trend analysis, correlation analysis, and contribution analysis methods to explore the change characteristics of DI in urban construction. The study focused on the DI changes before, during, and after the construction of Chenggong University Town. At the same time, combined with the data of IA, GDP, and POP, the influence intensity of human activities on the DI change was analyzed to provide theoretical support for the rational planning and development of the region. The conclusions are as follows:

1.  From 2001 to 2020, the summer DI of Chenggong District showed an upward trend, with a growth rate of 0.0563 °C/a. The area with a significant growth rate was relatively high. The growth rate in the western region was higher than that in the eastern region, especially in the northwest and central regions. In the pre-UT period, the growth rate of DI was low, and the expansion of high DI areas was slow. In the post-UT period, DI rose rapidly, and the growth rate was significantly higher than before the construction and the overall period. This showed that with the continuous construction and economic development of Chenggong District, problems such as the change in land use type and the increase of population have resulted in the change in the regional thermal environment, resulting in the decline of human comfort.

Especially after the construction of the UT in 2010, the influx of a large number of people and the establishment of University supporting facilities exacerbated the rising rate of DI. During UT, affected by the complex changes in land use types, the DI in Chenggong District showed fluctuations in time, but there was no obvious change in space.

2.  From the monthly changes, the DI of Chenggong District showed an upward trend in the three months of summer, with the fastest growth rate in June, and the area with the significant growth rate was the highest. In the pre-UT period, except for June, DI rose in other months, but the change rate was low, and the regional spatial change was not obvious. In the post-UT period, the DI in all months increased significantly compared with that before the construction, and all of them were higher than the overall increasing rate. The change rate in June was the highest compared with that before the construction. Meanwhile, compared with other areas, UT showed a significant upward trend in June, with the highest proportion of growth areas. This may be due to the change in the regional thermal environment caused by a large number of population movements during the summer vacation in UT in July and August, which reduced the growth rate of DI. This also proves that the construction of Chenggong University Town has deepened the interference intensity of the expansion of the new city on the regional thermal environment.

3.  The trend analysis of the DI showed that the development of the economy and society and the continuous strengthening of human activities had affected the human body's perceived comfort in the environment. Therefore, this study explored the depth of its impact. The results showed that the average DI in the development process of Chenggong District was significantly positively correlated with IA, GDP, and POP. Through the analysis of the contribution rate, it was found that the contribution intensity of IA to the DI change was 52.7%. From the monthly average, the correlation between DI and human activity factors in June was much higher than in other months. There was a significant positive correlation among all factors. This showed that human activities had a strong impact on DI.

The above research results proved the changes in human comfort during the development of Chenggong District, and found that the construction of University City had a strong impact on it. At the same time, human activities had a deep impact on DI. At present, with the development of urbanization and the construction of new policy-oriented cities, the increase of IA will have a strong impact on the regional thermal environment. Therefore, during the construction of new urban areas, attention should be paid to the impact of the increase in IA and population influx on human comfort. At the same time, buildings and infrastructure should be reasonably arranged to alleviate the negative effects of policy-based urban construction on human life and health.

**Author Contributions:** Conceptualization, K.Y. and Y.L.; Formal analysis, Y.Z., C.Z. and Z.P.; Writing—original draft, Y.Z. and Z.P.; Writing—review & editing, Y.Z., C.Z., K.Y., Z.P., L.T., H.D., C.W. and Y.L. All authors have read and agreed to the published version of the manuscript.

**Funding:** This work was supported by the National Natural Science Foundation of China (NSFC): Yi Luo 41761084 and the Ten Thousand Talent Plans for Young Top-notch Talents of Yunnan Province: Yi Luo YNWR-QNBJ-2019-200.

**Data Availability Statement:** The data that support the findings of this study are available from the corresponding author upon reasonable request.

**Acknowledgments:** The authors would like to thank all the authors for their hard work in this research.

**Conflicts of Interest:** The founding sponsors had no role in the design of the study, in the collection, analyses, or interpretation of data, in the writing of the manuscript, or in the decision to publish the results.

## Appendix A

Appendix A includes parameter information for thermal environmental monitoring stations.

**Table A1.** Thermal environment monitoring station sensor parameters: monitoring per minute.

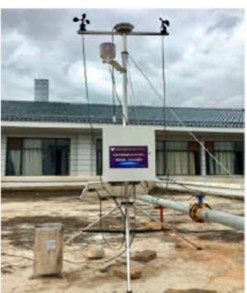

Monitoring Station 1

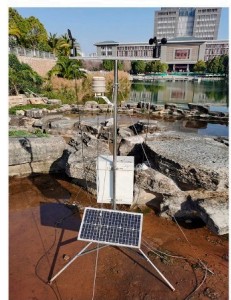

Monitoring Station 2

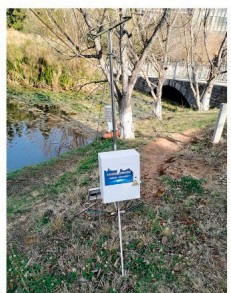

Monitoring Station 3

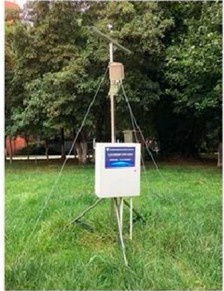

Monitoring Station 4

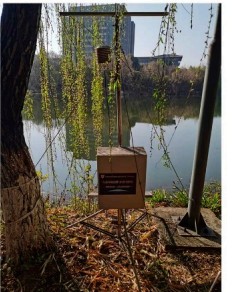

Monitoring Station 5

| Name | Measuring Range | Resolution Ratio | Error |
|---|---|---|---|
| Wind velocity sensor | 0–70 m/s | 0.1 m/s | $\pm(0.3 \pm 0.03v)$ m/s |
| Wind transducer sensor | 0–360° | 1° | ±3° |
| Temperature sensor | −50–+100 °C | 0.1 °C | ±0.4 °C |
| Air humidity sensor | 0–100% RH | 0.1% RH | ±3% |
| Barometric pressure sensor | 10–1100 hpa | 0.1 hpa | ±0.3 hpa |
| Rain sensor | 0–4 mm/min | 0.2 mm | ±4% |

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
