# Peer review of "Temporal and Spatial Effects of Urbanization on Regional Thermal Comfort"

_land, doi:10.3390/land11050688_

Round 1

Reviewer 1 Report

Dear authors,

I have read your work with interest. The results and conclusions are not in doubt. But it is necessary to improve the presentation of the results and the description of the methodology.

The main methodological remark on the article concerns correlation and regression analysis. You used Pearson's correlation coefficient. The Pearson correlation coefficient is used for the normal distribution of variables. Have you tested for normal distribution? To do this, you can use the Shapiro-Wilk test. If any of the variables has a non-normal distribution, then instead of the Pearson correlation coefficient, the Spearman correlation coefficient should be used.

The rest of the comments are about minor fixes and improvements:

1) Figure 1 needs a number of improvements:

a) Enlarge the figure (the whole figure 1 or only the bottom map) as much as the page margins allow. The bottom map (Chenggong district) has many small details. Enlarging the figure will allow you to see them better.

b) There are no geographical names on the lower map. Write the names of the two lakes (Dianchi and Yangzonghai) on the lower map. Use blue font for the lake names. Perhaps in this territory there are settlements or parts of the city, the names of which you consider important to sign. For such labels on the map use black font.

c) Unfortunate colors are chosen for the road and railways. Readers will associate blue with rivers and green with vegetation. Also, the green lines are hard to see against the background of green fills. I recommend using black for railroads. For roads, you can use a dark orange color. I recommend using black for railroads. For roads, you can use a dark orange color. Instead of an orange college town border, a dark red (as for Chenggong district) dashed line can be used.

d) Chose the wrong color for arable land. It's forest green, but darker. This looks illogical. Better make arable land in gray.

e) The map with the location of the city of Kunming does not show the meaning of the colors. Probably these colors indicate the relief (heights of the terrain). Place a scale on this card so readers can clearly understand the meaning of the colors. Instead of the green-yellow-purple scale, I suggest using the more traditional colors for the heights. For low altitudes - green. As the height increases, let the color change from green through yellow to dark orange or brown.

2) In Figure 5, there is a problem with the color scales. You use a separate color scale for each map. All scales have the same colors, but different ranges of values. Therefore, the same colors on different maps can mean different values. I propose a solution to this problem. Make one common scale for all maps. This scale should be discrete, with 5-7 gradations. Use the same colors: from red through yellow to blue. Also increase the size of Figure 5 as much as the page margins allow.

3) You use the letter r for Pearson's correlation coefficient in line 215. It is right. Further on, you use the letter P everywhere for Pearson's correlation coefficient. This is wrong. Use the traditional notation - letter r.

If you use the Spearman correlation coefficient, then it is denoted by the Greek letter rho (ρ).

4) The abbreviation VIF stands for variance inflation factor. And you incorrectly give the full name as variance expansion factor” (line 374). Write the correct meaning of the VIF abbreviation.

5) You wrote nowhere what software was used. Write what software is used for correlation and regression analysis. Also write what software you used to process MODIS products.

6) Line 134-135. What is the exact temperature? Write the temperature value.

7) Line 135-136. Write numerical values for sunshine (number of sunny days or duration of sunshine). If such information is not available, please let me know in your response.

8) Line 188. Please write in what coordinate system did you reproject the MOD11A2.

9) Line 194. In parentheses after the formula, explain that DN is a digital number. For specialists in remote sensing, this abbreviation is clear. But specialists from other fields may not know it.

10) There are doubts about the correctness of the name of subsection 2.3.3. It seems there is no analysis of the contribution of factors. But there is a correlation analysis and regression analysis. I would call it "2.3.3. Correlation analysis and regression analysis".

11) Figure 6 has illogical colors. It is better to use green to reduce (1 - dark green and 2 - light green), to increase, use red (4 - light shade, 5 - dark shade). Intermediate (3) - yellow.

Reviewer 2 Report

The manuscript presents the discomfort incurred by increased surface temperature in the Chenggong district of Yunnan province in China. The result of the research are important to the urban land managers in controlling the urban heat island created by increased urbanization and population density. The approach is quite simple and focus more on environmental issues rather than scientific originality. Still the research is of importance to the readers of the concerned field. However, there is still some space in the manuscript that can be improved prior its publication.

  1. Since the research is more concerned on the discomfort index induced by construction of university town (UT) i.e pre-UT and post-UT, I suggest to revise the title based on UT rather than urbanization as urbanization includes other parts in cities.
  2. Please arrange keywords alphabetically
  3. The introduction is written well, however, inclusion of the originality of the current research and how the research product will be utlilized by the concerned stakeholders is required. For consideration of discomfortness in south asian cities you may refer updated papers such as  Maharjan et al. (2021) In addition, why the particular research is required in the study region needs to be emphasized citing the existing research gaps
  4. Figure 1 Captions should be well explained in details and provide information about the base map used and use known legends (for ex. blue lines are used for rivers rather than roads). Better to remove second from the grid line.
  5. In study area, I recommend to discuss on the land use and land cover types too which has substantial role in controlling the surface temperature.
  6. Line 194: Please insert equation separately
  7. In methods sub section, authors explained about the data acquisition from ECMWF and equation of LST, but in the result part they directly jumped to discomfort index. Why is that? If the authors are excluding these contents better to explain with reasons otherwise I suggest to include in the result section as DI is associated with LST and RH.
  8.  In figure 5, better to use single legend classifying as described in Table 1 rather than for each years which makes confusion to readers on linkage of DI classification and spatio-temporal distribution. 
  9. Fig 5. Is there any meaning of coordinates provided? If possible provide the location of UT for post-UT period
  10. I wonder, how long was the construction period of UT and how did the author considered the construction period as it is neither pre-UT nor post-UT. During the construction period DI might have increased compared to pre-UT and post-UT.
  11. Line 407-408, compared with the abstract (line 23-27) aren't they contradictory?
  12. In discussion, line 471-472, does the vacation has such impact on the the correlation between DI and IA in a larger scale (district level)? Please consider.
  13. What about the colling agents in the study region such as open spaces, ponds, and trees? 

Round 2

Reviewer 2 Report

The authors have extensively revised the manuscript and now looks good for publication.